# Implementation of a Dimmable LED Driver with Extendable Parallel Structure and Capacitive Current Sharing

**Kuo-Ing Hwu** [1,*]🔘**, Yu-Kun Tai** [1] **and Hsiang-Hao Tu** [2]

[1]   Department of Electrical Engineering, National Taipei University of Technology, 1, Section 3, Zhongxiao East Road, Taipei 10608, Taiwan; ttmc@ms17.hinet.net
[2]   Delta Electronics Inc., Neihu, Taipei 11466, Taiwan; Patrick.tu@deltaww.com
*   Correspondence: eaglehwu@ntut.edu.tw; Tel.: +886-2-27712171 (ext. 2159)

**Abstract:** A dimmable LED driver along with an extendable series structure and interleaved capacitive current sharing is presented herein, the LED connection of which is changed from the traditional series structure to the proposed parallel structure. The number of LED strings can be extended. As the number of LED strings is increased, the output voltage of this LED driver and the voltage stress on the main switch are ideally not influenced. Moreover, only one current sensor is needed to achieve current control and dimming. In this paper, the basic operating principle of the proposed LED driver is described and analyzed. Finally, the effectiveness of this LED driver is demonstrated by experiment based on the field-programmable gate array (FPGA).

**Keywords:** capacitive current sharing; current-sharing interleaved capacitor; dimmable; extendable parallel structure; FPGA; LED

## 1. Introduction

As generally recognized, LEDs are getting more attractive in the world due to their small size, light weight, and long life [1,2]. An LED is driven by the current due to its behavior like a diode [3]. The higher the current in the LED, the higher is the forward-biased voltage across the LED [4]. Furthermore, the higher the temperature in the LED, the lower is the forward-biased voltage across the LED. In general, the arrangement of LEDs is first in series and then in parallel, so as to avoid a high voltage across the output of the LED driver. And hence, the current balance among LED strings is very important so as to avoid uneven currents in the LED strings. These uneven currents will affect the LED luminance and cause the temperature in the LED to be increased and the life of the LED to be reduced. Therefore, many literatures have presented current-sharing methods for LED strings [5–21] so as to make currents distributed among LED strings as identically as possible. The LED current-sharing methods are classified into two types. One is active [5–9], and the other is passive [10–21]. As for the active current-sharing method, each LED string contains one current regulator and one current sensor to balance currents among LED strings. The main demerit of this method is the high complexity of the circuit. Hence, the passive current sharing method is presented to overcome this disadvantage. This method can be classified into two types. One is based on the differential-mode transformer, and the other is based on the energy-transferring capacitor. For the first type to be considered [10–14], as the currents in two LED strings are not identical, this transformer will be activated, and hence, the two currents will be forced to be regulated as identically as possible. In [10], an LED driver is constructed of one Zeta converter with several current-balancing transformers such that the magnetizing energy can be recycled. In [11], several daisy-chained transformers are

applied to current-balance multiple LED strings. In [12], only for two LED strings to be considered, an LED driver is made of one traditional boost converter with one differential transformer used as a current-sharing device. In [13], the LED driver is the same as that shown in [11], except for isolation capacitors used in the former. In [14], an LED driver is established by one twin-bus converter along with several differential transformers such that the voltage stress on the switch is reduced. However, in practice, the current-sharing error comes from the magnetizing current of the transformer. For the second type to be considered [15–21], the current balance between the two LED strings is based on the ampere-second balance. In [15], an LED driver constructed of an isolated LCLC resonant circuit along with capacitive current balance. In [16], an LED driver is built from one traditional boost along with one coupled inductor and capacitive current balance such that the output voltage can be upgraded. In [17], an isolated CLL resonant converter with several balance capacitors is applied to driving multiple LED strings. In [18], isolated and non-isolated LED drivers are constructed by the traditional converters with switched capacitors used. In [19], an LED driver is constructed by one voltage-boosting converter with automatic current balance and zero-voltage switching such that the switching loss can be reduced. In [20], an LED driver is built from one traditional boost converter along with capacitive current balance, input ripple cancellation, and passive clamp such that the input current ripple is reduced and the voltage gain improved. In [21], a two-channel LED driver is established by one coupled-inductor converter along with capacitive current balance and one passive regenerative snubber such that the voltage gain is upgraded and the leakage energy can be recycled. However, in actuality, the current sharing error comes from the leakage current of the current-sharing capacitor.

Based on the aforementioned literatures, each circuit from [10–21] has the capability of extending the number of LED strings to two or more, if necessary, except for [12,20,21]. Since the current balance based on the differential transformer will occupy a relatively large space, the current balance of the proposed LED driver is based on the capacitor. The proposed LED driver is used to improve the LED driver in [18]. In [18], due to the LED strings connected in series, the voltage gain is the sum of all the voltages across the output capacitors as shown in (4) in [18] divided by the input voltage. Furthermore, the number of LED strings cannot be increased to more than four. To overcome this problem, by means of current-sharing interleaved capacitors, the voltage gain of the LED driver is not influenced at all as the number of LED strings is increased. Above all, the voltage stress on the main switch is always the same for any number of LED strings used.

## 2. Proposed LED Driver Circuit

Since the number of LED strings shown in [18] is four, in order to effectively describe the behavior of the current-sharing interleaved capacitors used in the proposed LED driver, the number of LED strings is set to six. The proposed LED driver circuit in Figure 1 is constructed of one switch, $Q_1$; one input inductor, $L$; five current-sharing interleaved capacitors, $C_1$, $C_2$, $C_3$, $C_4$, and $C_5$; six diodes, $D_1$, $D_2$, $D_3$, $D_4$, $D_5$, and $D_6$; and six output capacitors $C_{o1}$, $C_{o2}$, $C_{o3}$, $C_{o4}$, $C_{o5}$, and $C_{o6}$. In addition, six LED strings are used as load.

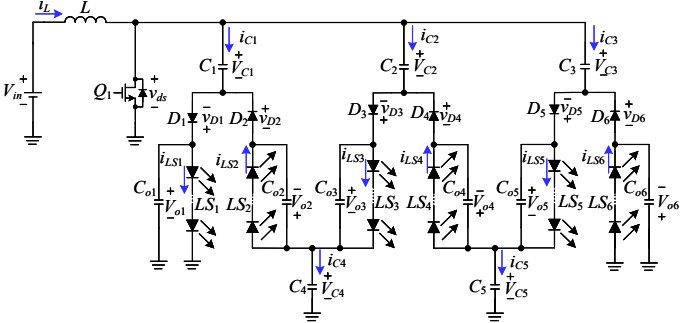

**Figure 1.** Proposed LED driver.

Prior to analysis of the basic operating principle of this circuit, some associated symbols and some assumptions are to be given in the following.

(1)　All the capacitors are large enough such that the voltages across them can be regarded as constant.
(2)　The input voltage is signified by $V_{in}$ and the output voltages are represented by $V_{o1}$, $V_{o2}$, $V_{o3}$, $V_{o4}$, $V_{o5}$, and $V_{o6}$.
(3)　The currents in $L$ and $Q_1$ are indicated by $i_L$ and $i_{ds}$, respectively.
(4)　The currents in $C_1$, $C_2$, $C_3$, $C_4$, and $C_5$ are denoted by $i_{C1}$, $i_{C2}$, $i_{C3}$, $i_{C4}$, and $i_{C5}$, respectively.
(5)　The currents in $C_{o1}$, $C_{o2}$, $C_{o3}$, $C_{o4}$, $C_{o5}$, and $C_{o6}$ are signified by $i_{o1}$, $i_{o2}$, $i_{o3}$, $i_{o4}$, $i_{o5}$, and $i_{o6}$, respectively.
(6)　The voltages on $L$ and $Q_1$ are represented by $v_L$ and $v_{ds}$, respectively.
(7)　The voltages on $C_1$, $C_2$, $C_3$, $C_4$, and $C_5$ are expressed by $V_{C1}$, $V_{C2}$, $V_{C3}$, $V_{C4}$, and $V_{C5}$, respectively.
(8)　The switching period is denoted by $T_s$.
(9)　The turn-on time interval of $Q_1$ is $DT_s$, where $D$ is the duty cycle.
(10)　The switch, the inductor and all diodes and capacitors are viewed as ideal, and all the LED strings are identical.
(11)　The gate driving signal for $Q_1$ is signified by $v_{gs}$.
(12)　The circuit is operated in the continuous conduction mode (CCM), that is, there are two operating states over one switching cycle, as shown in Figure 2.

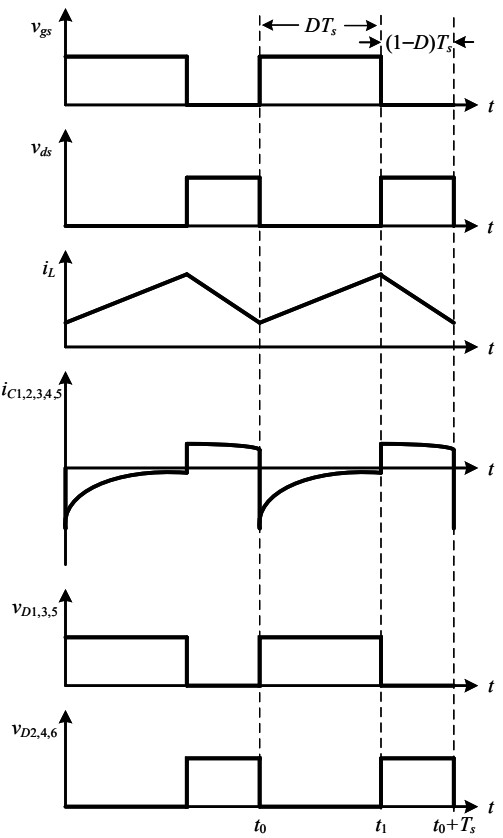

**Figure 2.** Proposed LED driver.

*2.1. Operating Principle Analysis*

(1)　State 1: [$t_0 \le t \le t_1$]: As shown in Figure 3, $Q_1$ is turned on. At the same time, $D_1$, $D_3$, and $D_5$ are turned off, whereas $D_2$, $D_4$, and $D_6$ are turned on. During this state, the voltage across $L$ is $V_{in}$,

$L$ is magnetized, whereas $C_1$, $C_2$, $C_3$, $C_4$, and $C_5$ are discharged through $Q_1$ and provide energy to $LS_2$, $LS_4$, and $LS_6$ and $C_{o2}$, $C_{o4}$, and $C_{o6}$. In addition, the energy required by $LS_1$, $LS_3$, and $LS_5$ is provided by $C_{o1}$, $C_{o3}$, and $C_{o5}$, respectively.

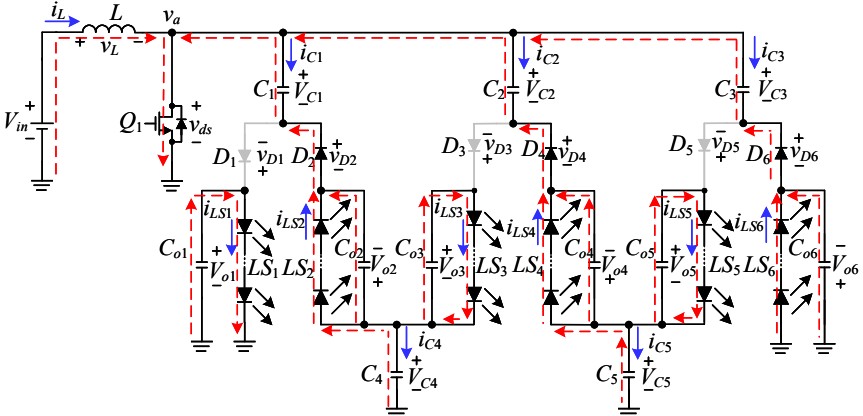

**Figure 3.** Circuit operating behavior of state 1.

(2) State 2: $[t_1 \leq t \leq t_2]$: As shown in Figure 4, $Q_1$ is turned off. At the same time, $D_2$, $D_4$, and $D_6$ are turned off, whereas $D_1$, $D_3$, and $D_5$ are turned on. During this state, the voltage across $L$ is $V_{in} - v_a$, where $v_a = V_{C1} + V_{o1}$. The inductor $L$, together with $V_{in}$, charges $C_1$, $C_2$, $C_3$, $C_4$, $C_5$, $C_{o1}$, $C_{o3}$, and $C_{o5}$, whereas the energy required by $LS_2$, $LS_4$, and $LS_6$ are provided by $C_{o2}$, $C_{o4}$, and $C_{o6}$, respectively.

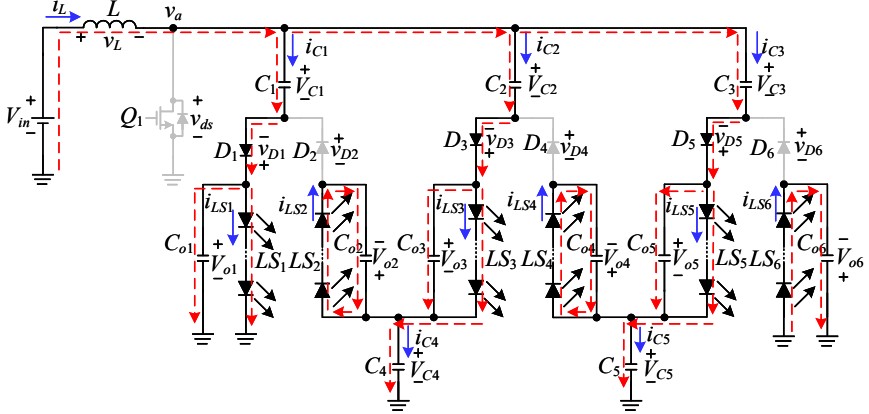

**Figure 4.** Circuit operating behavior of state 2.

### 2.2. Voltage Gain

As $Q_1$ is turned on, the voltage across $L$, $v_{L(on)}$, can be expressed as

$$v_{L(on)} = V_{in}. \tag{1}$$

As $Q_1$ is turned off, the voltage across $L$, $v_{L(off)}$, can be expressed as

$$v_{L(off)} = V_{in} - v_a. \tag{2}$$

According to the voltage-second balance over one switching cycle, the following equation can be obtained:

$$DV_{in} + (1 - D)(V_{in} - v_a) = 0. \tag{3}$$

By rearranging (3), the input voltage $V_{in}$ can be obtained as

$$V_{in} = v_a(1 - D). \tag{4}$$

During the turn-on period of $Q_1$, the following equations can be found:

$$\begin{cases} V_{C1} + V_{C4} - V_{o2} = 0 \\ V_{C2} + V_{C5} - V_{o4} = 0 \\ V_{C3} - V_{o6} = 0 \end{cases}. \tag{5}$$

During the turn-off period of $Q_1$, the following equations can be found:

$$\begin{cases} V_{C1} + V_{o1} = v_a \\ V_{C2} + V_{o3} + V_{C4} = v_a \\ V_{C3} + V_{o5} + V_{C5} = v_a \end{cases}. \tag{6}$$

By summing the equations shown in (5), the following equation can be obtained:

$$V_{C1} + V_{C2} + V_{C3} + V_{C4} + V_{C5} - V_{o2} - V_{o4} - V_{o6} = 0. \tag{7}$$

By summing the equations shown in (6), the following equation can be obtained:

$$V_{C1} + V_{C2} + V_{C3} + V_{C4} + V_{C5} + V_{o1} + V_{o3} + V_{o5} = 3v_a. \tag{8}$$

By subtracting (7) from (8), the voltage $v_a$ can be obtained as

$$v_a = \frac{1}{3}(V_{o1} + V_{o2} + V_{o3} + V_{o4} + V_{o5} + V_{o6}). \tag{9}$$

Finally, by substituting (9) into (4), the corresponding voltage gain can be found to be

$$\frac{v_a}{V_{in}} = \frac{\frac{1}{3}\sum\limits_{m=1}^{6} V_{om}}{V_{in}} = \frac{1}{1 - D}. \tag{10}$$

From (10), it can be seen that the required duty cycle is determined by averaging all the voltages across LED strings and then multiplying this result by two. If all the LED strings are identical, then (10) can be simplified to be

$$\frac{v_a}{V_{in}} = \frac{2V_{o1}}{V_{in}} = \frac{1}{1 - D}. \tag{11}$$

From (11), it can be seen that the duty cycle can be determined by the output voltage of one LED string. That is, the output voltage and duty cycle of the proposed LED driver are kept constant as the number of LED strings is increased, and Equation (11) still holds.

## 2.3. Boundary Condition of Input Inductor L

The boundary condition for $L$ is described as

$$\begin{cases} 2I_L \geq \Delta i_L \Rightarrow L \text{ works in the CCM} \\ 2I_L \leq \Delta i_L \Rightarrow L \text{ works in the DCM} \end{cases}, \tag{12}$$

where $I_L$ and $\Delta i_L$ are DC and AC components of $i_L$.

For analysis convenience, it is assumed that the input power is equal to the output power, and hence, the input current $I_{in}$ can be represented by

$$I_{in} = \frac{1}{1-D} \times \frac{V_o}{R_{eq}},$$　　　　　　　(13)

where $R_{eq}$ is the equivalent output resistance.

The average current of $i_L$, $I_L$, can be expressed as

$$I_L = I_{in}$$　　　　　　　(14)

$$I_L = \frac{1}{1-D} \times \frac{V_o}{R_{eq}}.$$　　　　　　　(15)

Furthermore, the current ripple of $i_L$, $\Delta i_L$, can be represented by

$$\Delta i_L = \frac{V_L \Delta t}{L} = \frac{V_{in} D T_s}{L}.$$　　　　　　　(16)

Therefore, as $2I_L \geq \Delta i_L$, the input inductor $L$ will operate in the CCM, and hence, the following equation can be obtained:

$$
\begin{aligned}
& 2I_L \geq \Delta i_L \\
& \Rightarrow 2 \times \frac{1}{1-D} \times \frac{V_o}{R_{eq}} \geq \frac{V_{in} D T_s}{L} \\
& \Rightarrow \frac{2L}{R_{eq} T_s} \geq \frac{V_{in} D(1-D)}{V_o} \\
& \Rightarrow \frac{2L}{R_{eq} T_s} \geq (1-D)^2 D \\
& \Rightarrow K_L \geq K_{crit\_L}(D)
\end{aligned}
$$　　　　　　　(17)

where $K_L = \frac{2L}{R_{eq} T_s}$ and $K_{crit\_L}(D) = (1-D)^2 D$.

From (17), it can be seen that as $K_L \geq K_{crit\_L}(D)$, the input inductor $L$ operates in the CCM; otherwise, in the DCM. Therefore, the operation boundary curve of $L$ can be drawn as shown in Figure 5.

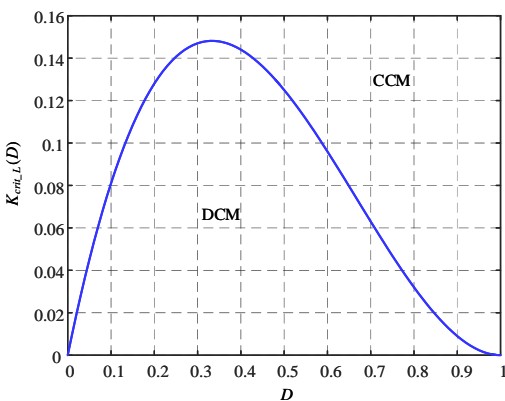

**Figure 5.** Operation boundary curve of the input inductor $L$.

## 2.4. Current Sharing Concept

For convenience of analysis, only the current-sharing interleaved capacitor $C_1$ is taken into account. According to the ampere-second balance, the absolute value of the electric charge during the turn-on period of $Q_1$, $Q_{C1\_on}$, is equal to the electric charge during the turn-off period of $Q_1$, $Q_{C1\_off}$; namely,

$$\left| Q_{C1\_off} \right| = Q_{C1\_on}$$　　　　　　　(18)

$$\frac{1}{T_s} \int_0^{DT_s} \left| i_{C1(on)} \right| dt = \frac{1}{T_s} \int_{DT_s}^{T_s} i_{C1(off)} \, dt. \tag{19}$$

The current in the LED string $LS_1$ is equal to the absolute average value of the negative part of $i_{C1}$ during the turn-on period of $Q_1$, and the current in the LED string $LS_2$ is equal to the average value of the positive part of $i_{C1}$ during the turn-off period of $Q_1$; therefore:

$$\left| I_{C1(on)} \right| = I_{C1(off)} = I_{LS1} = I_{LS2}. \tag{20}$$

## 3. System Control Strategy

Figure 6 shows the systems configuration of the proposed LED driver. This system contains the main power stage and the feedback control loop. The main power stage is constructed by the proposed LED driver. As for the feedback control loop, the current in the last LED string is sensed by the current sensor and then sent to the analog-to-digital converter (ADC), which transfers the analog signal to the digital signal. Afterwards, this digital signal is transferred to the field-programmable gate array (FPGA) to get a suitable control force after some calculations, so as to control the switch such that the currents in all the LED strings can be controlled near a desired value.

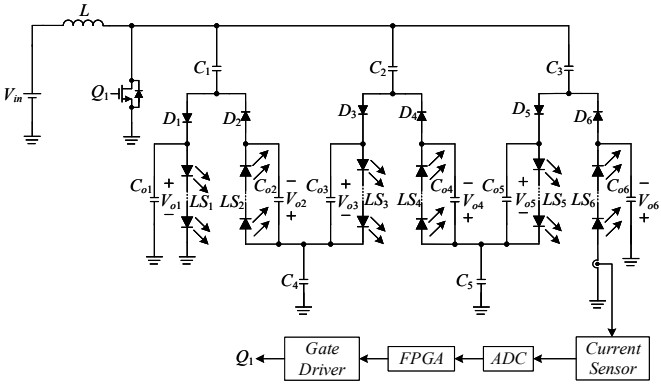

**Figure 6.** System block diagram of the proposed circuit.

## 4. Design Considerations

Prior to tackling this section, the system specifications are given in Table 1 as follows.

**Table 1.** System specifications.

| Circuit Operating Mode | CCM |
| --- | --- |
| Input voltage ($V_{in}$) | 12 ± 10% |
| Output voltage ($V_o$) | 27.6V($= 8 \times 3.45$V) |
| LED Rated current ($I_{o,rated}$)/Rated Power ($P_{o,rated}$) | 350 mA/29 W |
| LED Minimum current ($I_{o,min}$)/ Min. Power ($P_{o,min}$) | 87.5 mA/7.25 W |
| Switching frequency ($f_s$)/Period ($T_s$) | 100 kHz/10 µs |
| LED forward voltage ($V_F$) and current ($I_F$) at rated condition | 3.45 V/0.35 A |
| LED cut-in forward voltage ($V_{F,LED}$) at zero current | 2.7 V |
| LED strings | 6 strings with 4 LEDs per string |

### 4.1. Determination of Duty Cycle

According to the LED V-I curve shown in [22], the forward voltage $V_F$ and forward current $I_F$ at rated condition are 3.45V and 0.35A, respectively, whereas the cut-in forward voltage $V_{F,LED}$ at zero current is 2.7 V. Hence, the equivalent resistance $R_{F,LED}$ is 2.143 Ω.

### 4.1.1. Duty Cycles at Rated Load

The maximum duty cycle at rated load, $D_{max,rated}$, is determined by the minimum input voltage $V_{in,min}$ and is shown as follows:

$$D_{max,rated} = 1 - \frac{V_{in,min}}{I_{LED} \times \Sigma R_{F,LED} + \Sigma V_{F,LED}} = 1 - \frac{10.8}{0.35 \times 17.14 + 21.6} = 0.608 \tag{21}$$

The minimum duty cycle at rated load, $D_{min,rated}$, is determined by the maximum input voltage $V_{in,max}$ and is shown as follows:

$$D_{min,rated} = 1 - \frac{V_{in,max}}{I_{LED} \times \Sigma R_{F,LED} + \Sigma V_{F,LED}} = 1 - \frac{13.2}{0.35 \times 17.14 + 21.6} = 0.522 \tag{22}$$

### 4.1.2. Duty Cycles at Minimum Load

The maximum duty cycle at minimum load, $D_{max,min}$, is determined by the minimum input voltage $V_{in,min}$ and is shown as follows:

$$D_{max,min} = 1 - \frac{V_{in,min}}{25\% \times I_{LED} \times \Sigma R_{F,LED} + \Sigma V_{F,LED}} = 1 - \frac{10.8}{0.25 \times 0.35 \times 17.14 + 21.6} = 0.532 \tag{23}$$

Also, the minimum duty cycle at minimum load, $D_{min,min}$, is determined by the maximum input voltage $V_{in,max}$ and is shown as follows:

$$D_{min,min} = 1 - \frac{V_{in,max}}{25\% \times I_{LED} \times \Sigma R_{F,LED} + \Sigma V_{F,LED}} = 1 - \frac{13.2}{0.25 \times 0.35 \times 17.14 + 21.6} = 0.428 \tag{24}$$

### 4.2. Design of L

In order to make sure that the input inductor $L$ works in the CCM, the minimum average input inductor current $I_{L,min}$ should satisfy the following inequality:

$$I_{L,min} \geq \frac{\Delta i_L}{2}, \tag{25}$$

where $\Delta i_L$ is the input inductor current ripple.

Via setting the efficiency equal to one, the following equation can be obtained as

$$I_{L,min} = I_{in,min} = \frac{P_{o,min}}{V_{in,max}}. \tag{26}$$

Therefore, the inequality of $L$ can be expressed as

$$L \geq \frac{V_{in,max} D_{min,min} T_s}{2 \times I_{L,min}} = \frac{V_{in,max}^2 D_{min,min} T_s}{2 \times P_{o,min}} = \frac{13.2^2 \times 0.428 \times 10\mu}{2 \times 7.25} = 51.36\mu\text{H}. \tag{27}$$

Eventually, the value of $L$ is set at 60 µH.

### 4.3. Design of $C_{o1}$–$C_{o6}$

The values of $C_{o1}$–$C_{o6}$ are worked out at a rated condition. Since the voltages across the capacitors $C_{o1}$, $C_{o2}$, $C_{o3}$, $C_{o4}$, $C_{o5}$, and $C_{o6}$ are clamped by the identical LED strings, $V_{o1} = V_{o2} = V_{o3} = V_{o4} = V_{o5} = V_{o6}$. Furthermore, the capacitors $C_{o1}$, $C_{o3}$, and $C_{o5}$ are used to provide the energy for $L_{S1}$, $L_{S3}$, and $L_{S5}$ during the turn-on period of $Q_1$, respectively, whereas the capacitors $C_{o2}$, $C_{o4}$, and $C_{o6}$ are used to

provide the energy for $L_{S2}$, $L_{S4}$, and $L_{S6}$ during the turn-off period of $Q_1$, respectively. In addition, it is assumed that the voltage ripple for each output capacitor is set at 1% of its DC voltage. Therefore,

$$C_{o1} = C_{o3} = C_{o5} \geq I_{o,rated} \times \frac{D_{max,rated} \times T_s}{V_{o1} \times 1\%} = 0.35 \times \frac{0.608 \times 10\mu}{13.8 \times 0.01} = 15.4\mu F \tag{28}$$

$$C_{o2} = C_{o4} = C_{o6} \geq I_{o,rated} \times \frac{(1 - D_{max,rated}) \times T_s}{V_{o2} \times 1\%} = 0.35 \times \frac{(1 - 0.522) \times 10\mu}{13.8 \times 0.01} = 12.1\mu F. \tag{29}$$

Finally, one 22 µF/25 V capacitor is for $C_{o1}$ and also for $C_{o2}$, $C_{o3}$, $C_{o4}$, $C_{o5}$, and $C_{o6}$.

### 4.4. Design of $C_1$–$C_5$

The values of $C_1$–$C_5$ are figured out at rated condition. In this LED driver, the capacitors $C_1$, $C_2$, and $C_3$ have the same operating behavior, namely, $V_{C1} = V_{C2} = V_{C3}$, whereas the capacitors $C_4$ and $C_5$ have the same operating behavior, namely, $V_{C4} = V_{C5}$. From state 2 with the switch $Q_1$ being off and based on Kirchhoff's law (KVL), the following equation can be obtained:

$$- V_{C4} - V_{o3} - V_{C2} + V_{C1} + V_{o1} = 0 \tag{30}$$

Hence, $V_{C4} = 0$ since $V_{C1} = V_{C2}$ and $V_{o1} = V_{o3}$. In the same way, $V_{C5} = 0$.

From state 1 with the switch $Q_1$ being on and based on KVL, the following equation can be obtained:

$$V_{C1} - V_{o2} + V_{C4} = 0. \tag{31}$$

Hence, $V_{C1} = V_{o2}$ since $V_{C4} = 0$. In the same way, $V_{C2} = V_{o4}$ and $V_{C3} = V_{o6}$.

In addition, it is assumed that the voltage ripple for each capacitor is set at 1% of its DC voltage. Therefore,

$$C_1 = C_2 = C_3 \geq \frac{P_{o,rated}}{3 \times V_{in,min}} \times \frac{D_{max,rated} \times T_s}{V_{o1} \times 1\%} = \frac{29}{3 \times 10.8} \times \frac{0.608 \times 10\mu}{13.8 \times 0.01} = 39.1\mu F \tag{32}$$

Eventually, one 47 µF/25 V capacitor is for $C_1$ and also for $C_2$ and $C_3$. As for $C_4$ and $C_5$, they have the same capacitances as that for $C_1$ for design convenience.

In the following, the component specifications are tabulated in Table 2.

**Table 2.** Component specifications.

| Components | Specifications |
|---|---|
| MOSFET: $Q_1$ | IRF3250Z |
| Diodes: $D_1$, $D_2$, $D_3$, $D_4$, $D_5$, $D_6$ | STPS30L60C |
| Current Sharing Capacitors: $C_1$, $C_2$, $C_3$, $C_4$, $C_5$ | 22 µF/25 V |
| Output Capacitors: $C_{o1}$, $C_{o2}$, $C_{o3}$, $C_{o4}$, $C_{o5}$, $C_{o6}$ | 47 µF/25 V |
| Inductor | Core: T106-18B, $L = 60$ µH |
| Gate Driver | TC4420 |

## 5. Experimental Results

Figures 7–13 shows the waveforms related to the proposed LED driver at rated condition. Figure 7 shows the gate driving signal for $Q_1$, $v_{gs}$, the voltage on $Q_1$, $v_{ds}$, and the current in $L$. Figure 8 shows the gate driving signal for $Q_1$, $v_{gs}$, the voltages across $C_1$, $C_2$, and $C_3$, called $V_{C1}$, $V_{C2}$, and $V_{C3}$, respectively. Figure 9 shows the gate driving signal for $Q_1$, $v_{gs}$, and the voltages across $C_4$ and $C_5$, called $V_{C4}$ and $V_{C5}$, respectively. Figure 10 shows the gate driving signal for $Q_1$, $v_{gs}$, and the voltages across $D_1$, $D_2$, and $D_3$, called $v_{D1}$, $v_{D2}$, and $v_{D3}$, respectively. Figure 11 shows the gate driving signal for $Q_1$, $v_{gs}$, and the voltages across $D_4$, $D_5$, and $D_6$, called $v_{D4}$, $v_{D5}$, and $v_{D6}$, respectively. Figure 12 shows the gate driving signal for $Q_1$, $v_{gs}$, and the currents in $C_1$, $C_2$, and $C_3$, called $i_{C1}$, $i_{C2}$, and $i_{C3}$, respectively.

Figure 13 shows the gate driving signal for $Q_1$, $v_{gs}$, and the currents in $C_4$ and $C_5$, called $i_{C4}$ and $i_{C5}$, respectively. Figure 14 shows the voltage across $LS_1$, $V_{o1}$, the current in $LS_1$, $i_{LS1}$, the voltage across $LS_2$, $V_{o2}$, and the current in $LS_2$, $i_{LS2}$. Figure 15 shows the voltage across $LS_3$, $V_{o3}$, the current in $LS_3$, $i_{LS3}$, the voltage across $LS_4$, $V_{o4}$, and the current in $LS_4$, $i_{LS4}$. Figure 16 shows the voltage across $LS_5$, $V_{o5}$, the current in $LS_5$, $i_{LS5}$, the voltage across $LS_6$, $V_{o6}$, and the current in $LS_6$, $i_{LS6}$.

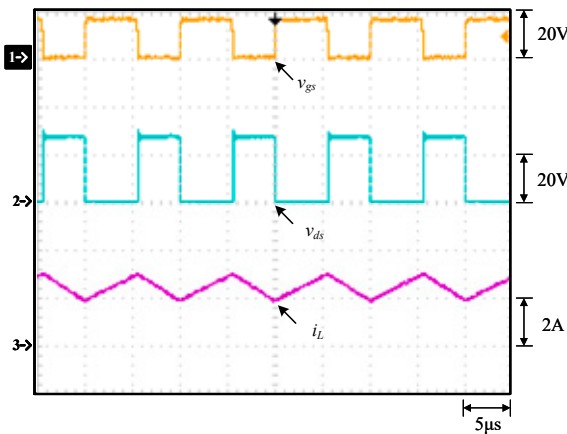

**Figure 7.** Measured waveforms at rated load: (1) $v_{gs}$; (2) $v_{ds}$; (3) $i_L$.

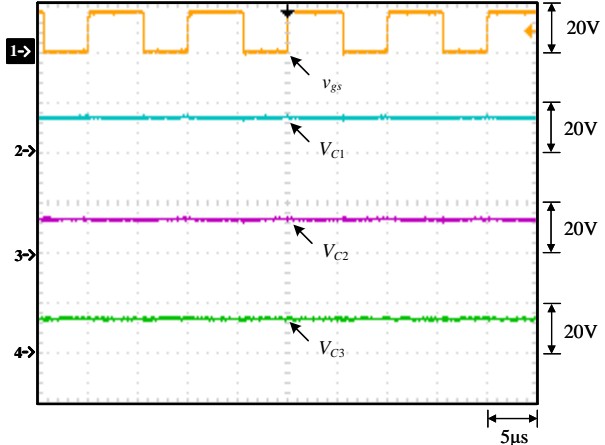

**Figure 8.** Measured waveforms at rated load: (1) $v_{gs}$; (2) $V_{C1}$; (3) $V_{C2}$; (4) $V_{C3}$.

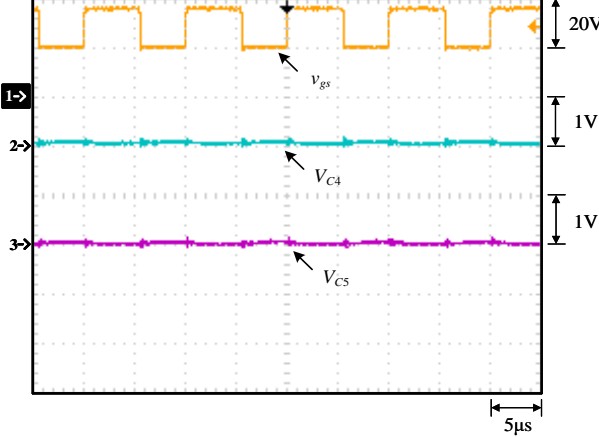

**Figure 9.** Measured waveforms at rated load: (1) $v_{gs}$; (2) $V_{C4}$; (3) $V_{C5}$.

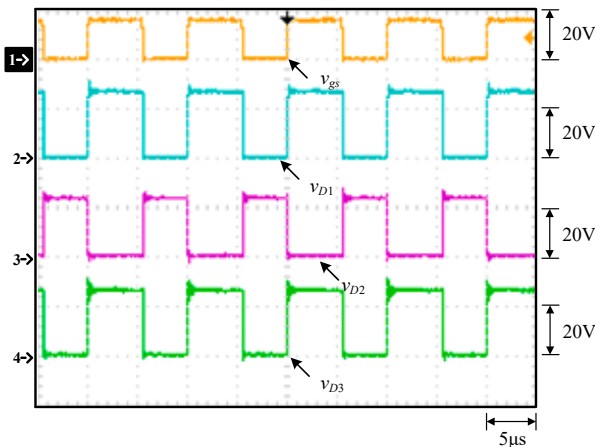

**Figure 10.** Measured waveforms at rated load: (1) $v_{gs}$; (2) $v_{D1}$; (3) $v_{D2}$; (4) $v_{D3}$.

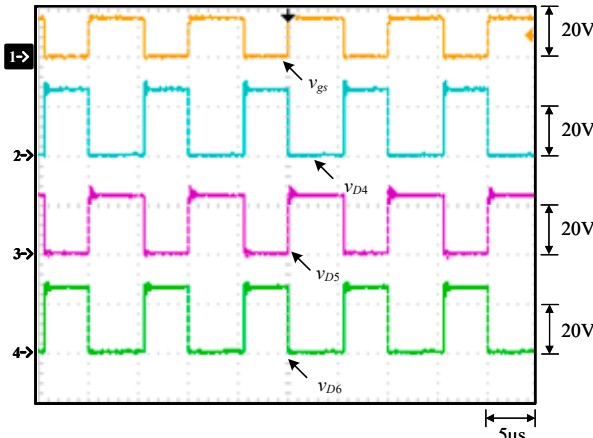

**Figure 11.** Measured waveforms at rated load: (1) $v_{gs}$; (2) $v_{D4}$; (3) $v_{D5}$; (4) $v_{D6}$.

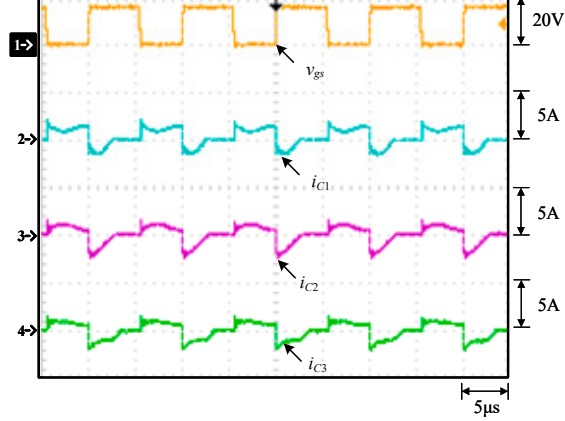

**Figure 12.** Measured waveforms at rated load: (1) $v_{gs}$; (2) $i_{C1}$; (3) $i_{C2}$; (4) $i_{C3}$.

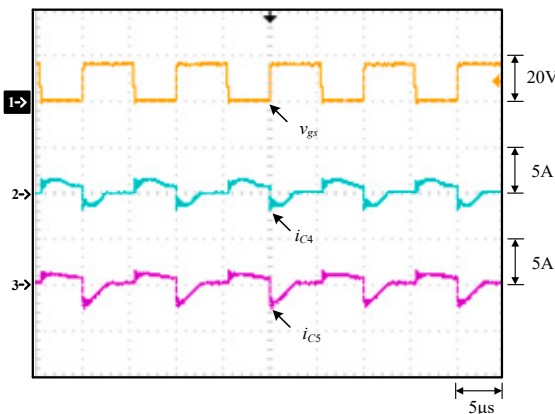

**Figure 13.** Measured waveforms at rated load: (1) $v_{gs}$; (2) $i_{C4}$; (3) $i_{C5}$.

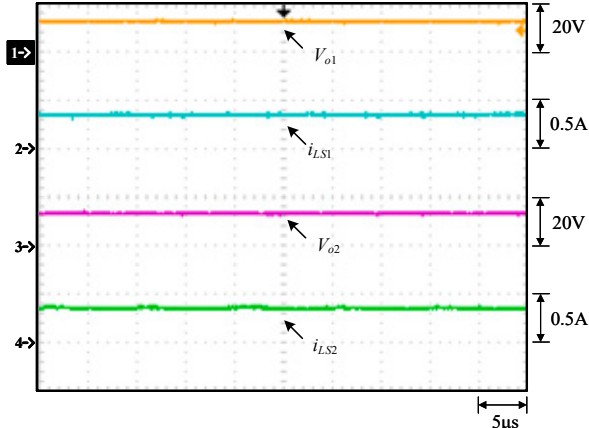

**Figure 14.** Measured waveforms at rated load: (1) $V_{o1}$; (2) $i_{LS1}$; (3) $V_{o2}$; (4) $i_{LS2}$.

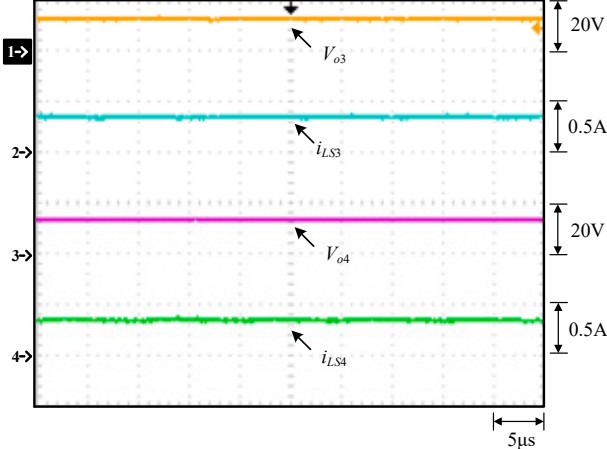

**Figure 15.** Measured waveforms at rated load: (1) $V_{o3}$; (2) $i_{LS3}$; (3) $V_{o4}$; (4) $i_{LS4}$.

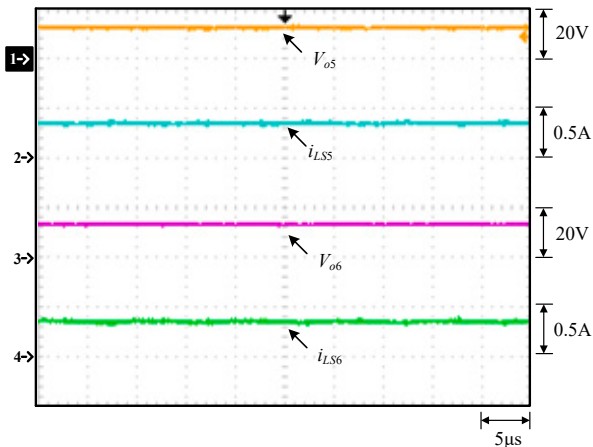

**Figure 16.** Measured waveforms at rated load: (1) $V_{o5}$; (2) $i_{LS5}$; (3) $V_{o6}$; (4) $i_{LS6}$.

From Figure 7, the input inductor $L$ operates in the CCM. From Figures 7, 10 and 11, the voltages across $Q_1$, $D_2$, $D_4$, and $D_6$ have voltage rings during the turn-off transient of $Q_1$, due to resonance among the body capacitance of $Q_1$, the diode parasitic capacitances of $D_2$, $D_4$, and $D_6$, and the line parasitic inductances. From Figure 8, the voltages across $C_1$, $C_2$, and $C_3$ are fixed at some values, whereas from Figure 9, the voltages across $C_4$ and $C_5$ are kept close to zero. From Figures 10 and 11, the voltages across $D_1$, $D_3$, and $D_5$ have voltage rings during the turn-on transient of $Q_1$, due to resonance among the diode parasitic capacitances of $D_1$, $D_3$, and $D_5$, and the line parasitic inductances. From Figures 12 and 13, the currents flowing through $C_1$ to $C_5$ have negative current rings since the voltages across $D_1$, $D_3$, and $D_5$ have voltage rings during the turn-on transient of $Q_1$, whereas the current flowing through $C_1$ to $C_5$ have positive current rings since the voltages across $D_2$, $D_4$, and $D_6$ have voltage rings during the turn-off transient of $Q_1$. From Figures 14–16, the current sharing between LEDs performs well, and the corresponding voltages across $LS_1$ to $LS_6$ are almost the same due to all LED strings being almost identical.

In addition, Tables 3–5 show the current sharing error percentage (CSEP) under an input voltage of 12 V and different current levels in LED strings. The definition of CSEP is shown as follows:

$$\text{CSEP}_y = \frac{I_{LSy} - \left( \sum_{x=1}^{m} I_{LSx} \right) \div m}{\left( \sum_{x=1}^{m} I_{LSx} \right) \div m} \times 100\% \tag{33}$$

where $\text{CSEP}_y$ is the CSEP of the $y$-th LED string, $m$ is the total number of LED strings, $I_{LSy}$ is the current flowing through the $y$-th LED string, and $\sum_{x=1}^{m} I_{LSx}$ is the sum of all the currents flowing through the LED strings.

Based on the calculated measurements from Tables 3–5, the CSEP values for any load locate between −1% and 1%. Therefore, the proposed LED driver has a good capability of current sharing.

**Table 3.** Associated current sharing error percentage (CSEP) measurements at minimum load.

|  | $LS_1$ | $LS_2$ | $LS_3$ | $LS_4$ | $LS_5$ | $LS_6$ |
|---|---|---|---|---|---|---|
| $I_{LS}$ (mA) | 85.4 | 86 | 85.2 | 86.6 | 85.3 | 86.5 |
| $V_{LED}$ (V) | 10.7 | 11.4 | 10.6 | 11.5 | 10.6 | 11.5 |
| Error (%) | −0.5 | 0.2 | −0.73 | 0.9 | −0.62 | 0.78 |

**Table 4.** Associated CSEP measurements at half load.

|  | $LS_1$ | $LS_2$ | $LS_3$ | $LS_4$ | $LS_5$ | $LS_6$ |
|---|---|---|---|---|---|---|
| $I_{LS}$ (mA) | 172 | 175 | 172 | 175 | 172 | 175 |
| $V_{LED}$ (V) | 11.3 | 12.1 | 11.2 | 12.1 | 11.2 | 12.2 |
| Error (%) | −0.86 | 0.86 | −0.86 | 0.86 | −0.86 | 0.86 |

**Table 5.** Associated CSEP measurements at rated load.

|  | $LS_1$ | $LS_2$ | $LS_3$ | $LS_4$ | $LS_5$ | $LS_6$ |
|---|---|---|---|---|---|---|
| $I_{LS}$ (mA) | 348 | 352 | 348 | 347 | 349 | 351 |
| $V_{LED}$ (V) | 12.4 | 13 | 12.3 | 13.2 | 12.3 | 13 |
| Error (%) | −0.33 | 0.81 | −0.33 | −0.62 | −0.04 | 0.53 |

In the following, how to measure the efficiency of the proposed LED driver is described. As shown in Figure 17, one current sensing resistor is in series with the input current path. First, a digital meter, called Fluke 8050A, is used to measure the voltage across this resistor so as to obtain the value of the input current. Sequentially, another digital meter, also called Fluke 8050A, is used to measure the input voltage. Accordingly, the input power can be worked out. As for the output power, six LED strings with the same number of LEDs, manufactured by Everlight Lighting Co., are used as the load of this LED driver. At the same time, the voltages across these six LED strings are measured by the other digital meter, also called Fluke 8050A. After this, the currents in these six LED strings are measured by a current probe, named TCPA 300, manufactured by Tektronix Co. Accordingly, the output power can be worked out. Finally, the corresponding efficiency can be figured out based on the calculated input and output powers. Figure 18 shows the curve of efficiency versus load current percentage. From this figure, it can be seen that the efficiency is above 90% all over the load range and can be up to 97.6%.

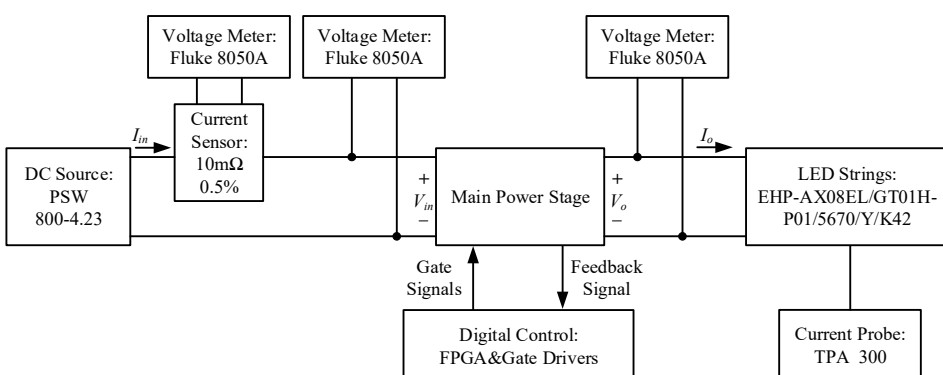

**Figure 17.** Block diagram for efficiency measurement.

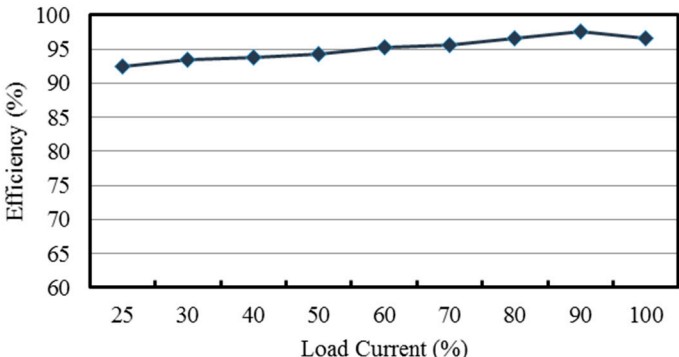

**Figure 18.** Curve of efficiency versus load current percentage.

## 6. Further Discussion on Current-Sharing Performance

In this section, to further demonstrate the current performance of the proposed LED driver, the number of LEDs for all the LED strings are not identical, e.g., three LEDs for $LS_1$, one LED for $LS_2$, three LEDs for $LS_3$, two LEDs for $LS_4$, four LEDs for $LS_5$, and three LEDs for $LS_6$. The waveforms shown in Figures 19–22, based on the PSIM software, are obtained under the rated LED current of 350 mA. Figure 19 shows the gate driving signal for $Q_1$, $v_{gs}$, the voltage on $Q_1$, $v_{ds}$, and the current in $L$, $i_L$. Figure 20 shows the voltage across $LS_1$, $V_{o1}$, the current in $LS_1$, $i_{LS1}$, the voltage across $LS_2$, $V_{o2}$, and the current in $LS_2$, $i_{LS2}$. Figure 21 shows the voltage across $LS_3$, $V_{o3}$, the current in $LS_3$, $i_{LS3}$, the voltage across $LS_4$, $V_{o4}$, and the current in $LS_4$, $i_{LS4}$. Figure 22 shows the voltage across $LS_5$, $V_{o5}$, the current in $LS_5$, $i_{LS5}$, the voltage across $LS_6$, $V_{o6}$, and the current in $LS_6$, $i_{LS6}$, since the voltage across each LED is assumed to be 3.45 V at rated current. Accordingly, based on (10), the calculated duty cycle, close to the simulated value shown in Figure 19, is

$$D = \frac{\frac{1}{3}\sum_{m=1}^{6} V_{om} - V_{in}}{\frac{1}{3}\sum_{m=1}^{6} V_{om}} = \frac{18.4 - 12}{18.4} \approx 0.35 \tag{34}$$

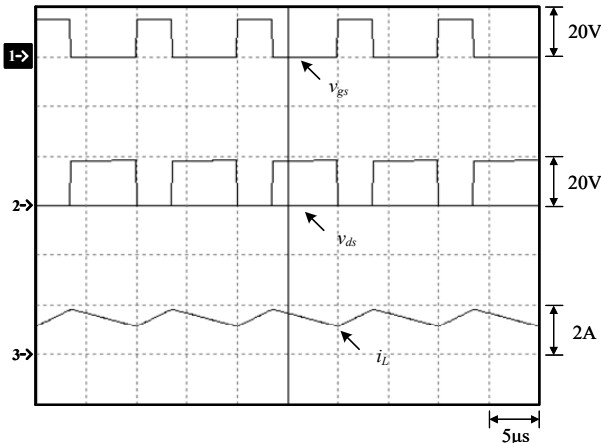

**Figure 19.** Simulated waveforms at rated load with LED counts being not all the same: (1) $v_{gs}$; (2) $v_{ds}$; (3) $i_L$.

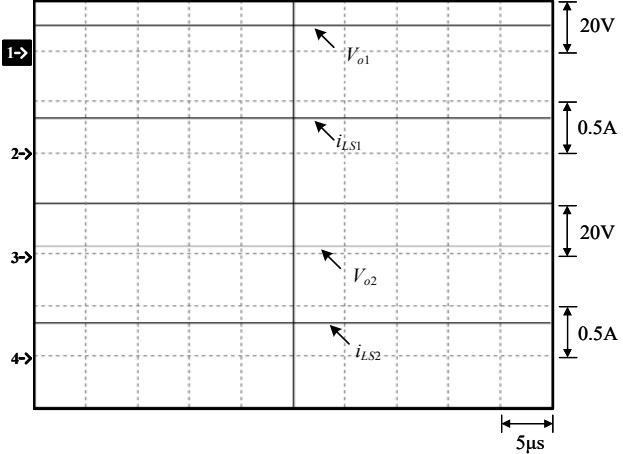

**Figure 20.** Simulated waveforms at rated load with LED counts being not all the same: (1) $V_{o1}$; (2) $i_{LS1}$; (3) $V_{o2}$; (4) $i_{LS2}$.

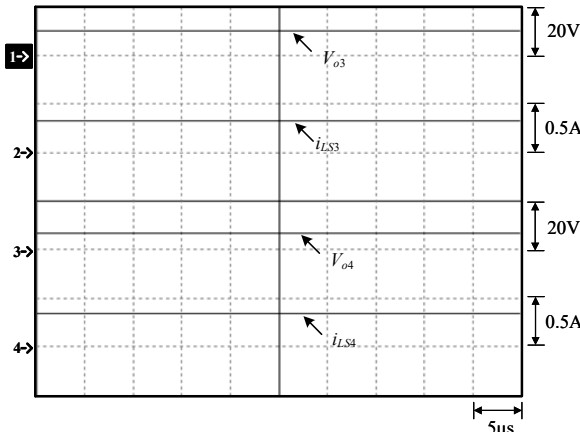

**Figure 21.** Simulated waveforms at rated load with LED counts being not all the same: (1) $V_{o3}$; (2) $i_{LS3}$; (3) $V_{o4}$; (4) $i_{LS4}$.

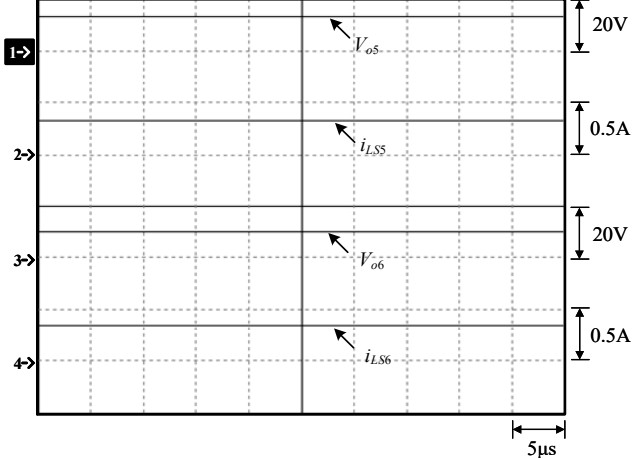

**Figure 22.** Simulated waveforms at rated load with LED counts being not all the same: (1) $V_{o5}$; (2) $i_{LS5}$; (3) $V_{o6}$; (4) $i_{LS6}$.

In addition, from Figures 19–22, it can be seen that even though the LED counts for individual LED strings are not the same, the currents flowing through all the LED strings are almost identical due to current control and current-sharing interleaved capacitors.

## 7. Conclusions

For the circuits shown in the literature [10–21], each circuit has the capability of extending the number of LED strings to two or more if necessary, except for [12,20,21]. Since the current balance based on the differential transformer will occupy a relatively large space, the current balance of the proposed LED driver is based on the capacitor. In this paper, an LED driver with extendable parallel structure and automatic current balance is presented. This LED driver is modified from series-type LED strings in [18] to parallel-type LED strings. The output voltage of the proposed LED driver with parallel-type LED strings is determined by averaging all the voltages across LED strings and then multiplying this result by two, different from the LED driver with series-type LED strings shown in [18], the output voltage of which is the sum of all the voltages across LED strings shown in (4) in [18]. Therefore, in [18], the voltage rating of the main switch will be increased, causing the turn-on resistance of the main switch to be increased. In addition, the number of LED strings cannot be increased to more than four. For the proposed LED driver, if all the LED strings are identical, then the output voltage is

kept constant as the number of LED strings is increased; if not, then the output voltage is changed slightly. Moreover, only one current sensor is required to realize current control and dimming.

**Author Contributions:** The conception was presented by K.-I.H., who also was responsible for editing this paper. Y.-K.T. surveyed the existing papers and wrote the software program. H.-H.T. carried out experimental setup and verification. K.-I.H. was in charge of project administration.

**Funding:** This research was funded by the Ministry of Science and Technology, Taiwan, under the Grant Number: MOST 108-2221-E-027-051.

**Acknowledgments:** The authors gratefully acknowledge the support of the Ministry of Science and Technology, Taiwan, under the Grant Number MOST 108-2221-E-027-051.

**Conflicts of Interest:** The authors declare no conflict of interest.

## Nomenclature

| | |
|---|---|
| $Q_1$ | Main switch |
| $C_1$, $C_2$, $C_3$, $C_4$, $C_5$ | Current-sharing interleaved capacitors |
| $L$ | Input inductor |
| $D_1$, $D_2$, $D_3$, $D_4$, $D_5$, $D_6$ | Diodes |
| $C_{o1}$, $C_{o2}$, $C_{o3}$, $C_{o4}$, $C_{o5}$, $C_{o6}$ | Output capacitors |
| $LS_1$, $LS_2$, $LS_3$, $LS_4$, $LS_5$, $LS_6$ | LED strings |
| $T_s$ | Switching period |
| $f_s$ | Switching frequency |
| $D$ | Duty cycle |
| $t_0$, $t_1$ | Time instants |
| $R_{eq}$ | Equivalent output resistance |
| $v_{gs}$ | Gate driving signal for $Q_1$ |
| $v_{ds}$, $v_a$ | Voltage across $Q_1$ |
| $i_{ds}$ | Current in $Q_1$ |
| $V_{C1}$, $V_{C2}$, $V_{C3}$, $V_{C4}$, $V_{C5}$ | Voltages across $C_1$, $C_2$, $C_3$, $C_4$, $C_5$ |
| $i_{C1}$, $i_{C2}$, $i_{C3}$, $i_{C4}$, $i_{C5}$ | Currents in $C_1$, $C_2$, $C_3$, $C_4$, $C_5$ |
| $i_{C1(on)}$ | Current $i_{C1}$ during turn-on period of $Q_1$ |
| $i_{C1(off)}$ | Current $i_{C1}$ during turn-off period of $Q_1$ |
| $I_{C1(on)}$ | Constant value of $i_{C1}$ during turn-on period |
| $I_{C1(off)}$ | Constant value of $i_{C1}$ during turn-off period |
| $v_L$ | Voltage across $L$ |
| $v_{L(on)}$ | Voltage across $L$ during turn-on period of $Q_1$ |
| $v_{L(off)}$ | Voltage across $L$ during turn-off period of $Q_1$ |
| $i_L$ | Current in $L$ |
| $I_{L,min}$ | Minimum DC value of $i_L$ |
| $\Delta i_L$ | Peak-to-peak value of current ripple of $i_L$ |
| $v_{D1}$, $v_{D2}$, $v_{D3}$, $v_{D4}$, $v_{D5}$, $v_{D6}$ | Voltages across $D_1$, $D_2$, $D_3$, $D_4$, $D_5$, $D_6$ |
| $i_{D1}$, $i_{D2}$, $i_{D3}$, $i_{D4}$, $i_{D5}$, $i_{D6}$ | Currents in $D_1$, $D_2$, $D_3$, $D_4$, $D_5$, $D_6$ |
| $i_{LS1}$, $i_{LS2}$, $i_{LS3}$, $i_{LS4}$, $i_{LS5}$, $i_{LS6}$ | Currents in $LS_1$, $LS_2$, $LS_3$, $LS_4$, $LS_5$, $LS_6$ |
| $I_{LS1}$, $I_{LS2}$ | Dc values of $i_{LS1}$, $i_{LS2}$ |
| $V_{in}$ | Input voltage |
| $V_{in,max}$ | Maximum input voltage |
| $V_{in,min}$ | Minimum input voltage |
| $V_o$ | Output voltage |
| $V_{o1}$, $V_{o2}$, $V_{o3}$, $V_{o4}$, $V_{o5}$, $V_{o6}$ | Voltages across $C_{o1}$, $C_{o2}$, $C_{o3}$, $C_{o4}$, $C_{o5}$, $C_{o6}$ |
| $i_{Co1}$, $i_{Co2}$, $i_{Co3}$, $i_{Co4}$, $i_{Co5}$, $i_{Co6}$ | Currents in $C_{o1}$, $C_{o2}$, $C_{o3}$, $C_{o4}$, $C_{o5}$, $C_{o6}$ |
| $I_{in}$ | Input current |
| $I_{in,min}$ | Minimum input current |
| $I_{o,rated}$ | LED rated current |
| $I_{o,min}$ | LED minimum current |

| $P_{o,min}$ | Minimum output power |
|---|---|
| $P_{o,rated}$ | Rated output power |
| $D_{max,rated}$ | Maximum duty cycle at rated load |
| $D_{min,rated}$ | Minimum duty cycle at rated load |
| $D_{max,min}$ | Maximum duty cycle at minimum load |
| $D_{min,min}$ | Minimum duty cycle at minimum load |
| $Q_{C1\_on}$ | Electric charge in $C_1$ during turn-on period |
| $Q_{C1\_off}$ | Electric charge in $C_1$ during turn-off period |
| $V_F$ | LED forward voltage |
| $V_{F,LED}$ | LED forward cut-in voltage |
| $R_{F,LED}$ | LED resistance |
| $I_{LED}$ | LED dc current |
| $I_{LSx}$ | Dc current in the *x-th* LED string |
| $I_{LSy}$ | Dc current in the *y-th* LED string |
| $CSEP_y$ | *y-th* $CSEP_y$ |

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
