# Peer review of "Implementation of a Dimmable LED Driver with Extendable Parallel Structure and Capacitive Current Sharing"

_applsci, doi:10.3390/app9235177_

Round 1

Reviewer 1 Report

The language is fluent, and it is easy to follow the content.

The paper technically sounds. However, the literature review is not complete, and it is not obvious what the novelty of this paper is. Also, in the result section, they could show how their method is an improvement against other methods that they had tried (which are available in the literature).

Author Response

Reponses to the Comments of Reviewer 1

The paper technically sounds. However, the literature review is not complete, and it is not obvious what the novelty of this paper is. Also, in the result section, they could show how their method is an improvement against other methods that they had tried (which are available in the literature).

Response:

Thanks a lot for Reviewer 1’s comments. The questions will be replied item by item as follows.

Q1: The literature review is not complete.

Response:

Please see black words on a yellow ground in the Introduction and References, in the revised paper.

Q2: It is not obvious what the novelty of this paper is.

Response:

Please see black words on a yellow ground on the last paragraph in the Introduction, in the revised paper.

Q3: In the result section, they could show how their method is an improvement against other methods that they had tried (which are available in the literature).

Response:

Please see black words on a yellow ground in the Conclusion, in the revised paper.

Reviewer 2 Report

This study demonstrated the fundamental background of LED driver operation and proposed the LED driver based-on consideration. This manuscript is well organized and contain clear conclusion. Reviewer recommend to publish as is.

Best regards

Author Response

Reponses to the Comments of Reviewer 2

This study demonstrated the fundamental background of LED driver operation and proposed the LED driver based-on consideration. This manuscript is well organized and contain clear conclusion. Reviewer recommend to publish as is.

Response:

Thanks a lot for Reviewer 2’s comments.

Reviewer 3 Report

The paper deals with the presentation of a dimmable LED driver with extendable series structure and capacitive current sharing. The work is well presented and supported by results. I suggest a minor revision before the publication. The abstract and the conclusion should included more results. Some references for the sentences in lines 24-28 should be added. Groups of references should be avoided by explaining better the contribution of each cited work (e.g. [6-10], [11-17], etc.). Even if the bullet point in section 2 describes many used symbols, a nomenclature should be added. The reason for the affirmation of the first sentence of section 2 ("In order to... six") should be explained better. 

Author Response

Reponses to the Comments of Reviewer 3

The paper deals with the presentation of a dimmable LED driver with extendable series structure and capacitive current sharing. The work is well presented and supported by results. I suggest a minor revision before the publication. The abstract and the conclusion should be included more results. Some references for the sentences in lines 24-28 should be added. Groups of references should be avoided by explaining better the contribution of each cited work (e.g. [6-10], [11-17], etc.). Even if the bullet point in section 2 describes many used symbols, a nomenclature should be added. The reason for the affirmation of the first sentence of section 2 ("In order to... six") should be explained better. 

Response:

Thanks a lot for Reviewer 3’s comments. The questions will be replied item by item as follows.

Q1: The abstract and the conclusion should be included more results.

Response:

Please see black words on a yellow ground in the Abstract and Conclusion, in the revised paper.

Q2: Some references for the sentences in lines 24-28 should be added.

Response:

Please see black words on a yellow ground on the first paragraph in the Introduction and four references [1]-[4] in the References, in the revised paper.

Q3: Groups of references should be avoided by explaining better the contribution of each cited work (e.g. [6-10], [11-17], etc.).

Response:

Please see more detailed descriptions for [6-10] and [11-17] in the Introduction, in the revised paper.

Q4: Even if the bullet point in section 2 describes many used symbols, a nomenclature should be added.

Response:

A nomenclature has been added to the revised paper.

Q5: The reason for the affirmation of the first sentence of section 2 ("In order to... six") should be explained better.

Response:

Please see black words on a yellow ground in the beginning of the first paragraph in section 2, in the revised paper.